# Challenges and Elements Needed for Children with Learning Disabilities in Teaching and Learning the Quran

**DOI:** 10.3390/children9101469

**Published:** 2022-09-26

**Authors:** Hafizhah Zulkifli, Syar Meeze Mohd Rashid, Suziyani Mohamed, Hasnah Toran, Norakyairee Mohd Raus, Mohd Nasri Suratman

**Affiliations:** 1Faculty of Education, Universiti Kebangsaan Malaysia, Bangi 43600, Malaysia; 2Faculty of Quranic and Sunnah, Universiti Sains Islam Malaysia, Nilai 71800, Malaysia; 3Yayasan Pendidikan Al-Quran Bagi Anak Istimewa, Nilai 71800, Malaysia

**Keywords:** challenges, elements, learning, children with learning disabilities, teaching, children education, learning styles

## Abstract

People with disabilities have the same right to access education and, therefore, the space and opportunity to study the Quran as other groups. However, there are some issues regarding the teaching of the Quran to students with special learning needs, such as from the aspects of the level of readiness of teachers, the mastery of Islamic Education among teachers who teach Special Education, and the use of teaching aids to teach the Quran. Therefore, this study aimed to identify the challenges, the need to develop a model of teaching and learning the Quran, elements to be included in the development of Quranic teaching, and a learning model for children with learning disabilities. A qualitative methodology was adopted using a case study design. The sample consisted of eight informants who volunteered to be involved in this research. The results show that there are seven main challenges in the teaching and learning of children with learning disabilities, i.e., a lack of stimulus materials, a lack of knowledge, limited time, uncontrolled behavior, traditional teaching, disabilities, and a lack of parental commitment. Thereafter, three themes arose in terms of identifying why it is necessary to develop Quran teaching and learning models for children with learning disabilities, such as the lack of an up-to-date model and the right to education. In addition, there were four themes concerning the elements that need to be included in the development of teaching and learning models for the Quran for children with learning disabilities, i.e., digital teaching aids, visual, audio, and kinesthetic learning styles, activities graded by the level of ability, and sensory support. It is hoped that this study will provide guidance to teachers to further strengthen the teaching profession pertaining to educating children with learning disabilities.

## 1. Introduction

The Quran is the greatest miracle revealed by Allah to the Prophet Muhammad through the mediation of the angel Gabriel. The revelation of the Quran is a guide for Muslims and human life because its content has been preserved throughout the ages and covers all aspects of human life. Every word and verse in the Quran contains guidelines, warnings, advice, messages, and teachings that must be emulated by all levels of society, especially Muslims. Thus, Quranic education is prominent in the life of every Muslim because Allah SWT established the constitution by which human beings must live in the Quran. This includes people with disabilities, who have the same right to access spaces and opportunities to learn the Quran [1], and rights to access education [2], as other typical groups. As Allah says in surah Al-Anbiya ‘107:
“And We have not sent you (O Muhammad), except as a mercy for all the worlds.”

This verse demonstrates that the Quran is revealed for everyone [3], including the disabled, who are not excluded from receiving Quranic education. The widest possible space and opportunity should be given to the disabled so that they are not behindhand in receiving Quranic education, which would result in their lives becoming unsteady due to the lack of a strong and incisive hold on the aspects of Quranic education.

However, there are certain issues related to Quranic education for the disabled, especially children with learning disabilities. These include the level of teacher readiness, the mastery of Islamic Education among teachers who teach special education, and the use of teaching aids. In addition to issues related to the development of a teaching and learning model of the Quran for children with special needs, there is also the issue of analyzing learning disabilities among children with such disabilities.

The first issue of teaching the Quran to children with learning disabilities is related to the level of readiness of teachers. Teachers need to be prepared in terms of knowledge, skills, attitudes, and teaching practices. However, the readiness of teachers is found to be at a moderate level, especially in teaching Jawi, Quran, Arab, and Fardhu Ain (J-QAF) to students with special needs [4]. Moreover, teachers who teach students with hearing problems are found to lack the skills to teach these children [5]. This is in line with the study of Ibrahim et al. [6], who stated that teachers are also unwilling to teach visually impaired students. In addition, teachers were found to lack the knowledge required for educating children with learning disabilities [4] and identifying the characteristics of children with learning disabilities [7].

Furthermore, another problem with teaching the Quran to children with learning disabilities is the mastery of Islamic Education teachers. The majority of teachers have not mastered and do not use appropriate methods for teaching children with learning disabilities [8]. This is because most teachers who teach Islamic Education for Special Education are not from the field of Special Education [4], meaning that they have no experience of training or teaching workshops for children with learning disabilities [9]. Although there are various methods to teach the Quran, including the use of the *Takrir* method to memorize Quranic verses for students with autism [10], the Fakih method for teaching deaf students, non-verbal students, and students with Down syndrome, the Abahata Al-Jabari method for teaching students with moderate disabilities, and the skin sensitivity and Braille methods for blind students [11], the method of teaching the Quran to disabled children with learning disabilities has received less attention.

In addition, teaching the Quran to children with learning disabilities should involve the use of teaching aids. A study by Ujang and Mohamad Salleh [4] stated that the use of teaching aids by teachers in special education schools is at an unsatisfactory level. The study of Abu and Saleh [12] also stated that one of the challenges of teaching the Quran to students with special needs is the constraints of teaching aids, the level of readiness of teachers, the teaching methods utilized by teachers, teachers’ knowledge, and limited time. Thus, teachers need to modify their teaching aids and integrate technology according to the needs of students with special needs [13].

The second issue is related to the Quran teaching and learning model for children with learning disabilities. A review of the literature determined that there are many studies on the development of modules for special needs education, such as the Health Education Webquests learning module for special needs students with learning disabilities [14] and the padlet module for deaf students [15]. However, there are only one or two studies on the teaching and learning model of the Quran for students with special needs in general [16] (Raus et al., 2017), and an even smaller number of studies on the teaching and learning model of the Quran for special education children with learning disabilities.

The third issue is related to the needs analysis of children with learning disabilities. Although the needs analysis of children with learning disabilities is a focus of previous studies, most of the studies involved the analysis of the sexual needs [17] and critical thinking of students with special learning disabilities [18], as well as the analysis of the training needs of special education senior assistant teachers [19]. By contrast, the Quran teaching and learning needs of children with learning disabilities have not been widely studied.

The current study is beneficial for teachers to understand the challenges and needs of teaching and learning for children with learning disabilities to improve the quality of teaching of special education teachers, particularly teachers who teach Islamic Education. This study aimed to identify challenges in the teaching and learning of children with learning disabilities, develop a model of teaching and learning the Quran for children with learning disabilities, and propose elements that need to be included in the development of a Quranic teaching and learning model for children with learning disabilities.

## 2. Literature Research

The success and sustainability of an education system depend on its curriculum design, which is based on the needs of individuals and society in achieving a set objectives and goals [19,20]. Therefore, an effective curriculum model should have a dynamic design that constantly seeks to expand and change in order to improve the quality of teaching or learning in a country [21]. As a developed model should provide guidance to users in the implementation of teaching and learning activities, the model must be developed based on the current needs of users to ensure that it can really be implemented properly. Therefore, there is a need for a model that is built based on the needs of its users to improve the quality of education in a country [19,21].

### 2.1. Students with Special Needs

According to the people with disabilities Act 2008 [22], people with disabilities are those with long-term physical, mental, intellectual, or sensory impairments that, when combined with various barriers, restrict their ability to fully and effectively participate in society. Disabled people are the minority group who need the same protection and assistance as Indigenous people, women, and children [23]. The United Nations initiated the Convention on the Rights of Persons with Disabilities to ensure that people with disabilities have equal rights to enjoy life.

In Malaysia, it is estimated that 1% of the population has special educational needs, whereas the global average is 10%. This percentage is below the actual number of students who have special educational needs [24]. The initial definition of a child by UNICEF is from the womb to the age of 8, while the Children Act 2001 defines a child as an individual under the age of 18. The Individuals with Disabilities Education Act (IDEA) states that children with special educational needs are children with autism, intellectual disability, hearing impairment including deafness, speech impairment, visual impairment, serious emotional disorders, orthopedic disabilities, late development, traumatic brain injury, other health problems, and specific learning disabilities. This group is eligible to receive special education and related services [25,26]. The People with Disabilities Act lists seven categories, namely, hearing impairment, visual impairment, speech impairment, physical disability, learning disability, mental disability, and various disabilities [22].

In the field of education, the term ‘people with disabilities’ refers to students with special educational needs. Students with special educational needs are a minority group that needs the attention and assistance of various parties, especially the local community, to ensure their survival through physical or spiritual access [26,27,28,29]. In fact, their teaching and learning requires specific techniques and methods to suit their needs and abilities [26,27,29]. Accordingly, the Ministry of Education Malaysia under The Special Education Division divides students with special educational needs into three categories, namely, students with hearing impairments, students with visual impairments, and students with learning disabilities.

### 2.2. Quranic Education for the Students with Special Educational Needs in Malaysia

Quranic education includes *Qiraat* recitation education, memorization methods, translation, tafsir studies, law studies, and so on [16]. In the current study, Quranic education for the disabled refers to the holistic exploration of the model and curriculum of Quran competency for disabled people, which is comprehensive from various aspects, whether offering or providing inclusive services suitable as a guide at all times and a blessing for the whole world (*rahmatan lil ‘Alamin*) [16].

In Malaysia, Quranic education is taught in schools through Islamic Education subjects. Under the Education Act 1996, Islamic Education must be taught in all primary or secondary schools with at least five Muslim students [30]. Therefore, all Muslim students, including students with special needs, must learn Islamic Education, which includes the study of the Quran. Nevertheless, teaching the Quran is difficult for teachers who teach students with various disabilities, including hearing, visual, and physical disabilities [30,31].

The content of the Islamic Education curriculum is implemented in Malaysian schools, especially at the primary level, as outlined by the Ministry of Education Malaysia in 2002. The Standard Curriculum for Special Education Primary Schools is developed in accordance with the National Education Philosophy, which is founded on the principles of an integrated approach, individual holistic development, equal educational opportunities and quality for all students, and lifelong education [31]. The curriculum standard of Islamic Education comprises seven main areas, namely, the Quran, Hadith, Faith, Worship, *Sirah,* Adab, and Jawi. The field of al-Quran is divided into three, namely, reading the verses of the Quran correctly and fluently, memorizing certain verses of the Quran correctly and fluently, and understanding the meaning of certain surahs and appreciating their teachings [32]. The Malaysian Ministry of Education through the Islamic Education Division (BPI) periodically reviews the Islamic Education curriculum to improve weaknesses, build suitability in terms of the implementation aspects with current needs, develop the ability to build students’ personality, and instill the seeds of strong and solid faith in facing various forms of current challenges [16].

In order to teach the Quran, teachers who teach Islamic Education need to master various skills appropriate to the learning needs of students according to their disability categories [33]. However, there are certain issues and challenges in the teaching of special education programs among teachers in Malaysian Special Education Schools. These include the academic and professional qualifications of teachers teaching students with special needs, knowledge of the characteristics of students with special needs, the ability and confidence of the teachers to deliver the content of Islamic Education to students with special needs [34], and the infrastructure that supports the teaching and learning of Islamic Education for students with special needs [35].

Quranic education must be implemented accordingly to students with special educational needs. This is because its implementation becomes a demand of duty to all. Students with special educational needs need to be given the space and opportunity to enjoy the benefits of a religious life. Therefore, empowerment in terms of the needs of the model in accordance with the differences in the ability of the students with special educational needs is actually based on the philosophy of the dignity of the position of the children of the Prophet Adam and the concept of effort orientation, which includes strategies, approaches, methods, and techniques of teaching and learning (PdP) [31,36]. Dynamic Quran competency is seen to be able to achieve the goal of holistic Quranic education in accordance with the demands of religion in order to convey the knowledge in the best way based on the level of ability.

## 3. Materials and Methods

### 3.1. Research Design

This research employed a qualitative research approach with a case study research design, which emphasizes the study of people’s conscious experience of their daily life and social action [37]. The qualitative research approach was appropriate to attain the research objective, which was to explore teachers’ experience in teaching the Quran to children with learning disabilities, including challenges that teachers face. It was also necessary to identify why the model is required and to understand the elements that need to be included in the model.

### 3.2. Setting and Participants

A non-probability sampling strategy was used in this research, with an initial purposeful sampling technique applied in the process of participant selection. This technique was chosen to make sure that the participants were qualified to provide rich information about their experience of teaching the Quran to students with a learning disability [38]. To ensure the qualification of the participants, a two-criterion-based selection method was established, namely, (i) teachers who were actively working in government schools or private special education centers, and (ii) teachers with at least five years of experience teaching students with a learning disability to recite the Quran. Later, network sampling strategies were administered to locate and identify participants that met the established criterion-based selection. In the end, a total of eight participants participated in this research. Table 1 provides participant information.

### 3.3. Data Collection

A semi-structured interview protocol was used to gather information. All the interviews were recorded using MP4, and field notes were made. Later, the interviews were transcribed verbatim using NVIVO software.

### 3.4. Data Analysis

The data were analyzed using a thematic analysis, as suggested by Glaser and Strauss [39]. The NVIVO software was used for this purpose. The first step of the current study was theme construction. The data were assigned to their respective themes using open coding. Then, the themes were named based on the data assigned to them.

### 3.5. Credibility, Consistency, and Transferability

Credibility, consistency, and transferability are the crucial aspects of qualitative research. These aspects demonstrate the veracity of the findings, the consistency of the result with the findings, and the transferability of research. Three techniques were employed in the current study to ensure the study’s credibility, consistency, and transferability: (1) member checks; (2) peer review; and (3) triangulation.

Member checks were conducted by handing over the verbatim transcripts to the respective participants. The participants were given two weeks to carefully read the verbatim transcripts. They were allowed to ask questions if they had a query about the content of the verbatim transcript. All the participants in the current research agreed to the content of the verbatim transcript and signed the consent form.

Next, a peer-review process was conducted by holding a focus group discussion with three experts. The experts were appointed to examine the themes that were developed. The experts also scrutinized the appropriateness of each interview quote mapped to its respective theme. On the basis of the focus group discussion, one subtheme was transformed to a theme. Later, a triangulation process was carried out by comparing the responses collected from the participants.

## 4. Findings

The findings of the current study are divided into three parts. Part A focuses on the challenges in teaching and learning of children with learning disabilities. Part B addresses the reason a teaching and learning model of the Quran for children with learning disabilities is needed. Lastly, Part C outlines the elements that need to be included in the development of a teaching and learning model of the Quran for children with learning disabilities.

### 4.1. Challenges in Teaching and Learning of the Quran for Children with Learning Disabilities

From the conducted interviews, there were seven challenges that occurred in the teaching and learning of the Quran for children with learning disabilities: a lack of stimulus materials, a lack of knowledge, limited time, uncontrolled behavior, traditional teaching, disabilities, and lack of parental commitment.

#### 4.1.1. Lack of Stimulus Materials

According to informants 1, 2, and 3, there was insufficient stimulus material to teach students with special needs if everything was used at the same time. For example, informant 1 stated: “*Well, among the problems that I identified in teaching Al-Quran for children with learning disabilities is lack of stimulus materials*”. Then, Informant 2 supported informant 1 by adding that there were not enough materials to aid in teaching in the following excerpt: *“In addition, there are also*
*challenges where teaching aids are not enough for all students. Some classes are conducted continuously, so other classes cannot use that stimulus material if the other class did not finish yet, so the use of teaching aids is not enough, which makes it difficult to teach the students.”* Informant 2 added that teaching aids were usually created by teachers. Therefore, if teachers lacked ideas, there were fewer teaching aids for students with special needs. The informant explained: *‘Most of the teaching aids are issued by the teachers. “So, after the teaching session, when we are fatigue, we will not have energy to brainstorm and have less ideas, less exposure, less ways to approach, and less ways to assess the needs of the children.”*

#### 4.1.2. Lack of Knowledge

The second challenge identified was a lack of knowledge. Most of the informants, namely, informants 1, 2, 3, and 8, desired knowledge to teach the Quran to students with special needs. For instance, informant 2 said: *“But the challenge is still the lack of knowledge”.*

The lack of knowledge includes knowledge about the disabilities and about the emotional management of children with learning disabilities. For example, informant 8 described a lack of knowledge when dealing with students with autism. She explained: *“I have very little knowledge about autism. The student came and we accepted him, if the students tantrum, i do not know how to handle him, because we had lack of knowledge in managing tantrum behavior, so the teacher needs a lot of knowledge”.* Next, informant 1 also stated: *“[…] based on my teaching experience, although we have been teaching for a long time, each student is different with different emotions and problems that we cannot anticipate. So, among the constraints is knowledge. You have to always add knowledge.”*

#### 4.1.3. Time

Furthermore, challenges also result from limited time, especially due to the schedule of teaching and learning of the Quran to children with learning disabilities. Informant 2 stated that the subject of the Quran was only taught in the evening. As a result, students were drowsy and lacking in enthusiasm. He said: “*Quran subject is done in the evening. Some of them attend schools in the morning, and only go to KLink Care Center in the evening where they have already felt tired and sleepy.”* In addition, the subject of Quran was not taught every day, causing students to easily forget and take the subject lightly, as stated by informant 2: “*In my opinion, some of the problems that can be identified is when the subject is not done every day, hence there is no daily repetition. It is quite difficult for both students and teachers.”*

#### 4.1.4. Uncontrolled Behavior

Informant 1 admitted that, based on his experience teaching students with special needs, students sometimes bring unresolved problems from home to the school. Therefore, teachers had to calm the student first before they started their teaching sessions. He stated: *“Okay, the problem that I identified when I taught children with special needs, is their behaviours. […] sometimes these children have an unresolved issue at home, but the issue is brought to school. For example, he did not get what he wanted at home, so when he attended the FAQEH Foundation (care centre), the issue at home affected his learning process in the classroom…. the children would throw a tantrum to get into his classroom and cause problems.”* This uncontrollable behavior did not only happen to school-aged students with special needs, but also among students aged 18 years and above. He said: *“This problem does not only involve small students because there are also students of mine who are 20 years old with behaviours still unresolved.”* Some instances of uncontrollable behaviors included tearing the Quran, chatting, and refusing to use the Quran to study it. According to informant 8: “*I have a student who when he doesn’t want to learn would tear the Quran. There are also times when he invites teachers to chat first until they are comfortable to learn Quran.”*

#### 4.1.5. Traditional Teaching 

Informant 2 also added that students with special needs were not comfortable using the traditional Quran to study the Quran. This is demonstrated in the words of informant 2: *“There are also those who do not want to use the Quran. They want to use a book with a larger script and colours. Some even bring their Quran because they do not want to use the traditional Quran.”* Moreover, some students were not interested in learning the Quran at all, as stated by informant 2: “*There are also some students who are not interested in learning the Quran and do not focus in the class, they want larger script.”* Informant 1 added that there were also students who refused to and were not interested in learning the Quran: “*There are also students who take almost half a year to like and want to learn the Quran.”*

#### 4.1.6. Disability 

Furthermore, some of the students forget what they learn. Informant 4 stated that children with learning disabilities were unable to retain information for longer. 

“*These boys are easy to forget. The challenge is, for example, we have been teaching for almost five or more months, then a school holiday started. A week’s leave wasn’t long at all. But when they returned to school after that, we had to go teach the same content again.*” Informant 4 added that the students were more forgetful when there was a long public holiday: “*Even though they are 18 years old, they cannot remember to read the words such as alif, ba and ta. If there is school holiday, they will forget what I teach and I need to teach again*”. In addition, informant 8 admitted that they were not able to listen to the digital pen used to learn the Quran. Therefore, he had to recite the Quran to the students. “*There are also students who can’t hear the pen. We can’t slow down or strengthen the digital pen. So, we read to them. There are also those who do not want to use the Quran.*” Next, according to informant 6, they were also unable to focus on the teaching and learning of the Quran: “*We are also faced with the problem of students who cannot sit for long periods of time in class. Some could last five minutes. Then they would want to play and so on.*”

#### 4.1.7. Lack of Parents Commitment

Another challenge associated with the teaching and learning of the Quran for children with learning disabilities is the lack of parental commitment. Parents play a role in educating children with special needs to read the Quran, but the current study found that there was no continuity between learning the Quran at school and at home due to various constraints. For example, according to informant 4: *“We do tell parents to connect and teach their children at home. But yeah, parents also work, right? So, I don’t want to say that there is no parental support, but this constraint prevents these children from continuing learning the Quran.”* According to informant 1, this causes students with special needs to miss Quranic lessons: *“Next, when there is no continuity of learning at home … he has become accustomed to not learning the Quran at home. This affects the learning of the Quran at school because there is no continuity of learning the Quran at home.”* In addition, parents were also unable to provide teaching aids such as digital Qurans at home as students used digital Qurans at school. This caused students not to study at home. Informant 3 explained: *“We use a digital Quran in class, but they are not using the same Quran at their home. Therefore, it is difficult for parents to apply the technique of teaching Quran at home because there are not enough teaching aids.”* Informant 4 added that internet access is necessary to succeed in online learning: “*At home […] needs to have the internet, as well as computer, laptop, or mobile phone. If we don’t have the necessities for online classes, students will have problems to learn.”*

### 4.2. Reason to Develop a Model of Teaching and Learning Quran

#### 4.2.1. Lack of Model

All informants agreed on the need to develop a model for teaching and learning the Quran for children with learning disabilities. Among them, informant 1 stated: *“In my opinion, there is a real need for a Quran learning model for children with learning disabilities.”* Informant 4 also agreed on the need for the development of a Quran model as stated: *“It is very necessary. To say it’s compulsory, it’s fardhu ain. It’s not fardhu kifayah. It is very much encouraged.”* This is because, in the opinion of informant 4, there is no Quran teaching model for students with special needs: *“I think there are not many Quranic teaching models in Malaysia especially to children with learning disabilities”*. Informant 3 agreed with this statement, explaining that there was no special module for the subject of the Quran: *“In addition, we also do not have a special module for the subject of the Quran. A module may help teachers to know the ways and steps they need when teaching Quranic subjects.”*

#### 4.2.2. Up to Date

The second theme for the development of a teaching and learning model of the Quran for children with learning disabilities is being up to date. The developed models must accord with the current needs. Informant 3 explained: *“Okay, the importance of a Quranic teaching model for the disabled is because we are the way people learn the Quran is changing. As, we really need to keep up with the current trend of modernity, the model of teaching the Quran needs changes. If one model is developed today, it may be irrelevant after ten years.”*

#### 4.2.3. Right to Education

The third theme for the development of teaching and learning models of the Quran for the disabled is the right to education. The right to education involves the opportunity given to students with special needs to learn the Quran. This was explained by informants 5 and 1: *“It is so important that more children with learning disabilities get the opportunity to learn and know the Quran.”* In addition: *”A model of teaching the Quran for children with learning disabilities is very important to be developed because the Quran is revealed to people of all groups.”* Informant 2 added that children with learning disabilities also have the right to know the holy verses of the Quran: *“For me, it is very important because they also have the right to know and recite the Quran, as well as its holy verses.”* Informant 4 also added that the opportunity for learning the Quran should not be limited to a typical group only. He explained: *“The Quran for all, the Quran for Muslims. The space and opportunity to study the Quran cannot be said to be limited to normal students only. We must adopt the approach that the Quran is for everyone. Allah accepts much, Allah accepts little. So, with the existence of this learning model, it can help at least a few children with disabilities who may have been very far from knowing Allah to get closer to Allah.”*

### 4.3. Elements Need to Be Included in the Model of Teaching and Learning Quran

The elements suggested by the informants of the current study to be included in the development of a teaching and learning model of the Quran for Children with learning disabilities were as follows: digital stimulus materials, visual, audio, and kinesthetic visual learning styles (VAK), activities graded by levels of ability, and sensory support.

#### 4.3.1. Digital Teaching Aids

Informant 3 suggested that the necessary element in the development of a Quran teaching and learning model for special needs students is digital materials. Informant 3 stated that if digital teaching aids are included in a model of teaching and learning development of the Quran for students with special needs, it will be easy to disseminate. He explained: “*Well, for me, it is very important that digitalisation be included in a Quran learning model for the disabled with learning disabilities […] The reason I suggest digitalising material is because with the sophistication of today’s modern equipment, such as laptops and smartphones, they can do all sorts of things. If you want to colour, draw, and communicate it with everyone, you can use the technology. We can change the way our children learn Quran by exposing them to various technological materials. The reason is that this technology is easier to spread compared to books and papers that are easily lost and damaged. This is what we need to emphasise now.”*

#### 4.3.2. Visual, Audio, and Kinesthetic Learning Styles

In addition, according to informant 1, the elements of visual, audio, kinesthetic learning style should be applied in the model: *“For elements or criteria that need to be included in a Quran teaching model, the first is visual methods, the second is auditory, the third is kinesthetic,... In short, all these elements are indeed suitable for these children.”*

#### 4.3.3. Activities by Level of Ability

Next, informants 2 and 4 expressed their support for the inclusion of activity elements graded according to the level of ability of students with special needs for the model: *“In my opinion, a model of teaching the Quran should have activities or techniques that are suitable for students and also activities that are according to the level of the students.”* Informant 4 said that: *“In my opinion, the elements or criteria that need to be presented in a model of teaching the Quran is that it can be applied to all groups with disabilities, not burdensome, easy to understand, and according to the students’ levels.”* This was also acknowledged by informant 6, who stated: *“This model needs to be developed. But it is best to try to celebrate the strengths and abilities of these special children. We know that learning disabilities like autism exist on a large spectrum, and that their abilities and capabilities are varied. So, I also need to know the level of ability of students.”*

#### 4.3.4. Sensory Support

Informant 5 added that the element that needs to be included is a sensory element. He explained: “*Before deciding on the elements to be included in a model of learning the Quran, first, we look at sensory problems, gross motor skills, fine motor skills, focus, and emotional control to identify how to attract students to participate in the Quran learning activity. This is important because these are present in children with learning disabilities. For instance, if they lack gross motor, we should think of what to do to help them focus on learning the Quran. Another example, when there are students who find it difficult to sit and refuse to pay attention when studying the Al-Quran, we should think of the activities to improve their attentive skills. So, identifying sensory elements is important for us to know so, it will ease the process of teaching children with learning disabilities in Quran.*”

## 5. Discussion

In conclusion, the results show that there are seven challenges in the teaching and learning of children with learning disabilities, namely, a lack of stimulus materials, a lack of knowledge, limited time, uncontrolled behavior, traditional teaching, disability, and a lack of parental commitment. Next, three themes arose in terms of identifying why it is necessary to develop Quran teaching and learning models for children with learning disabilities: the lack of model, the need to be up to date, and the right to education. In addition, there were four themes in terms of identifying the elements that need to be included in the development of teaching and learning models for the Quran for children with learning disabilities, namely, digital teaching aids, visual, audio and kinesthetic learning styles, activities graded by the level of ability, and sensory support.

Findings around objective one support to notion that the challenges in teaching and learning of the Quran for children with learning disabilities stem from teachers, children with learning disabilities, and parents. The findings of the current study are in line with those of the study of McKenzie et al. [40], who stated that families and teachers or parents need to collaborate to support the learning of students with special needs, especially those with learning disabilities. The findings of their study also revealed five themes, namely, communication, learning extension from school to home, support from families and teachers, dynamics and advocacy, and commitment from teachers and parents. 

Moreover, the current study supports the study by Abu Hamour [41], which reveals that almost half of students with learning disabilities in Jordan face behavioral problems. The study by Lewis et al. [42] used positive behavioral support (PBS) as a core intervention model to reduce behavioral concerns and improve the quality of life and occupational involvement of individuals with a learning disability who present challenging behaviors. The study looks at pre- and post-outcome measures that show a significant improvement in behavior, quality of life, and occupational outcomes for individuals with a learning disability presenting complex behavioral concerns.

Findings around objective 2 suggest that there is a need for the development of a Quran learning model for individuals with disabilities. Three factors were identified based on the thematic analysis of the current study. They are (i) the limited culturally appropriate Quran learning model in Malaysia, (ii) the evolution of time, and (iii) the right of individuals with disabilities to access education. As culture plays a crucial role in everyday life, especially when it comes to religious and spiritual education, an effective Quranic learning model needs to be designed and developed in accordance with Malaysian culture and style of life. 

Current features need to be identified and included in a learning model for individuals with disabilities to improve Quranic teaching. This can be achieved by considering the evolution of the technology, as well as new teaching approaches, strategies, and techniques. In addition, the shift of the special education paradigm from the welfare model to the social model needs to be considered. The social model considers that an individual with special needs has a right to access education [43], whereas the welfare model offers individuals with disabilities education because of sympathy. Therefore, accessible and appropriate education needs to be provided for them to access education. 

Nonetheless, there were some limitations in conducting this study. The first limitation is related to the fact that the research samples only included teachers who taught students with learning disabilities and did not cover teachers teaching students with autism and other disabilities. The second limitation was that this study used qualitative research to explore the needs analysis. Therefore, future research should use instruments such as survey research to cater to the problems and challenges faced by teachers teaching the Quran to students with special needs.

## 6. Suggestions

From this research, there are several suggestions for improving the teaching and learning of the Quran among children with learning disabilities. First, there must be cooperation between teachers and parents in teaching the Quran. For example, teachers at school used a digital electronic Quran to teach children with learning disabilities, so parents must have a digital electronic Quran to help children with learning disabilities to do revisions at home. Secondly, parents and teachers also have to participate in training or workshops on how to manage children with learning disabilities and how to teach the Quran either in the short term or in the long term. Such workshops are important because parents and teachers can learn how to solve problems associated with children with learning disabilities and parents help teachers by teaching the Quran at home. Finally, a collaboration with external parties such as non-government agencies to raise awareness among the community and help to promote the right to read the Quran among children with learning disabilities is required. 

## 7. Conclusions

In conclusion, a model of teaching and learning the Quran to children with learning disabilities is needed to overcome the problems in the teaching and learning of children with learning disabilities. Teachers teaching children with learning disabilities need a model that corresponds to the level of ability of the students. Following the current COVID-19 pandemic, teaching and learning must be asynchronous and aligned with asynchronous learning materials that meet academic standards to attend to the needs of children with learning disabilities [44]. 

The novelty of this study is that it is the first step toward increasing our knowledge around teaching and learning the Quran for children with learning disabilities, as prior research claimed that teachers lack the necessary knowledge, especially in terms of identifying the characteristics of children with learning disabilities, but also in terms of teaching methods. To fill the gap in this knowledge, this study suggests the elements needed to develop a model as a guide for Quran teachers to teach children with learning disabilities, namely, digital teaching aids, visual, audio and kinesthetic learning style, activities graded by level of ability, and sensory support.

This research adds new challenges for teaching and learning the Quran, such as uncontrolled behavior, disability, and a lack of parental commitment, compared to previous research that only stated challenges around constraints in terms of teaching aids, in addition to the level of readiness of teachers, the teaching methods of teachers, teachers’ knowledge, and limited time. 

This study advances the knowledge in the field of Quranic teaching to children with learning disability, as before this, there was no concern regarding reading the Quran for children with learning disabilities. Moreover, there is currently no guidance on how to teach children with learning disabilities.

## Figures and Tables

**Table 1 children-09-01469-t001:** Participant information.

Code of Participants	Experiences	Position	Higher Education	Place of Work
Informant 1	5 years	Teacher	Degree-Universiti Sains Islam Malaysia Master-Universiti Kebangsaan Malaysia	Foundation of al-Quran for Special Needs, No.51 Jalan Kenanga, 1/3 Laman Kenanga, Nilai Impian, 71800, Nilai, Negeri Sembilan Darul Khusus.
Informant 2	5 years	Teacher	Degree-Universiti Sains Islam Malaysia	K-Link Care Center, No. 2, Jalan BP3A, Taman Bukit Permata, 68100, Batu Caves, Selangor Darul Ehsan.
Informant 3	6 years	Operations Manager	Degree-Universiti Teknologi Mara	Pondok OKU, No. 18 Jalan Lavenda 1, Laman Lavenda, Nilai Impian, 71800, Nilai, Negeri Sembilan Darul Khusus.
Informant 4	12 years	Headmaster	Degree-Asia E Universiti	Sekolah Agama Menengah Pendidikan Khas, D/A JKR 5739 Sekolah Kebangsaan Pendidikan Khas, Johor
Informant 5	5 years	Teacher	Degree-Universiti Sains Islam Malaysia	No.20 & 22, Jalan Kompleks Niaga Vista Kirana 1, 75450 Bukit Katil, Melaka.
Informant 6	6 years	Teacher	Universiti Sains Islam Malaysia	No. AC 1/15-A, Blok B Jalan Plumbum AC7/AC Seksyen 7 Shah Alam.
Informant 7	5 years	Teacher	Degree-University of South Australia	No. AC 1/15-A, Blok B Jalan Plumbum AC7/AC Seksyen 7 Shah Alam.
Informant 8	6 years	Teacher	Degere-Universiti Sains Islam Malaysia	No.22 Jalan BM 7/19, Bandar Bukit Mahkota, Seksyen 7, 43000 Kajang, Selangor.

## Data Availability

Not applicable.

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
