# Peer review of "Challenges and Elements Needed for Children with Learning Disabilities in Teaching and Learning the Quran"

_children, 2022, doi:10.3390/children9101469_

Round 1

Reviewer 1 Report

1. Add participants profile in the methodology.

2. Insert a visual diagram to represent the narrative description of the literature, research findings and discussion.

2. What are/is the novelty of this research? Elaborate more. 

1 would suggest the topic : Challenges and Elements Needs to Children with Learning  Disabilities in  Malaysian Quranic Teaching and Learning  

Author Response

i had done the correction. pls see the attachment. thank you

Reviewer 2 Report

This study examined an issue rarely been examined before. The study is a first step to increase the knowledge of teaching and learning the Quran in children with learning disabilities.

I would like to propose some comments for the authors to improve their manuscript.

Title: “teaching and learning the Quran” should be added into the title.

Abstract:

1.          Full spelling of “VAK” should be added.

2.          How this study was conducted should be described briefly.

Introduction

1.          Full spellings of “SWT” and “SAW” should be added.

2.          I am not familiar with the words and language used to teach the Quran in Malaysia. Did it increase the difficulties to teach the Quran?

Discussion

1.          The limitations should be moved to Discussion section.

2.          The authors may add a section to propose their suggestions for improving teaching and learning the Quran based on the results of their study.
